# Dietary Diversity and Dietary Patterns in School-Aged Children in Western Kenya: A Latent Class Analysis

**DOI:** 10.3390/ijerph19159130

**Published:** 2022-07-26

**Authors:** Tiange Liu, Sherryl Broverman, Eve S. Puffer, Daniel A. Zaltz, Andrew L. Thorne-Lyman, Sara E. Benjamin-Neelon

**Affiliations:** 1Department of Health, Behavior and Society, Johns Hopkins Bloomberg School of Public Health, Baltimore, MD 21205, USA; dzaltz1@jhu.edu (D.A.Z.); sara.neelon@jhu.edu (S.E.B.-N.); 2Department of Biology, Duke University, Durham, NC 27708, USA; sbrover@duke.edu; 3Duke Global Health Institute, Duke University, Durham, NC 27710, USA; eve.puffer@duke.edu; 4Department of Psychology and Neuroscience, Duke University, Durham, NC 27708, USA; 5Center for Human Nutrition, Department of International Health, Johns Hopkins Bloomberg School of Public Health, Baltimore, MD 21205, USA; athorne1@jhu.edu

**Keywords:** micronutrient, dietary diversity, dietary pattern, children, Kenya, rural

## Abstract

Inadequate diet among children has both immediate and long-term negative health impacts, but little is known about dietary diversity and dietary patterns of school-aged children in rural Kenya. We assessed dietary diversity and identified dietary patterns in school-aged children in Western Kenya using a latent class approach. We collected dietary intake using a 24 h dietary recall among students in elementary schools in two rural villages (hereafter village A and B) in Western Kenya in 2013. The mean (SD) age was 11.6 (2.2) years in village A (*n* = 759) and 12.6 (2.2) years in village B (*n* = 1143). We evaluated dietary diversity using the 10-food-group-based women’s dietary diversity score (WDDS) and found a mean (SD) WDDS of 4.1 (1.4) in village A and 2.6 (0.9) in village B. We identified three distinct dietary patterns in each village using latent class analysis. In both villages, the most diverse pattern (28.5% in A and 57.8% in B) had high consumption of grains, white roots and tubers, and plantains; dairy; meat, poultry, and fish; and other vegetables. Despite variation for some children, dietary diversity was relatively low for children overall, supporting the need for additional resources to improve the overall diet of children in western Kenya.

## 1. Introduction

Inadequate dietary intake among school-aged children in Kenya may have both immediate and long-term negative health impacts, including delayed physical growth and impaired cognitive development in childhood [1] as well as an increased risk of cardiometabolic diseases in adulthood [2]. Dietary diversity is a key indicator of high-quality diet and is of particular interest to school-aged children with high nutrient requirements for growth. Low dietary diversity, characterized by inadequate quantities and unbalanced distribution of food groups, often results in nutritional deficiencies, especially in micronutrients [3]. Studies conducted among children in Kenya have traditionally focused on children aged five years and younger, given well-established diet-disease pathways related to dietary intake in very young children [4]. External events, such as violence, emergencies, floods, and famines in Kenya underscore the importance of observing and intervening to improve dietary intake among school-aged children. Targeting school-aged children provides an opportunity to not only enhance growth and nutritional status but also eliminate risks of long-term diseases prior to adulthood.

The women’s dietary diversity score (WDDS) is a simple tool to quantify dietary diversity and reflect micronutrient adequacy at the population level. It was developed by the Food and Agriculture Organization (FAO) of the United Nations and can be used with a single 24 h dietary recall [5]. The 10-food-group-based WDDS has outperformed other dietary diversity assessments that were based on either 7, 9, or 12 food groups in reflecting micronutrient adequacy [5,6,7]. However, no prior studies have used the 10-food-group-based WDDS to evaluate dietary diversity in school-aged children in Kenya. Previous studies have used dietary diversity scores other than the 10-food-group-based WDDS and reported low scores in young children in Kenya [8,9,10,11] and school-aged children in other African settings [12,13,14,15]. Therefore, there is a need to examine dietary diversity using the 10-food-group-based WDDS in school-aged Kenyan children.

Additionally, actual dietary patterns in school-aged children in Kenya are unclear. A more traditional diet in Kenya is based on grains and cereals, with additions of vegetables, fruits, and tubers when available. However, dietary intake varies depending on culture, geographic location, and socioeconomic status [16], and also changes with urbanization [17,18], modernizing food systems [4,19], and climate [20]. A growing body of literature has assessed dietary intake in different populations in Kenya [21,22,23,24,25,26,27]. These studies have reported consumption frequency or prevalence of selected foods or food groups, but did not identify or characterize dietary patterns. They also did not target school-aged children, who may have different dietary needs and patterns from both younger children and adults [28]. Recently, the latent class analysis approach has been increasingly used in low- and middle-income settings to characterize dietary patterns based on existing food consumption [29,30]. The approach is ideally suited for identifying underlying subgroups in a population characterized by distinct features, such as dietary intake profiles. Yet, it has not been applied in school-aged Kenyan children to assess their dietary patterns. In this study, we aimed to address these research gaps by (1) assessing dietary diversity using the 10-food-group-based WDDS and (2) identifying dietary patterns using latent class analysis in a sample of school-aged children from two rural villages in western Kenya. 

## 2. Materials and Methods

### 2.1. Study Population

We collected data from a convenience sample of children in elementary schools from 2 rural villages (referred to as village A and B) on the shores of Lake Victoria in western Kenya (Migori County, previously part of Nyanza Province) in 2013. We used data collected for baseline assessments for a larger study evaluating a natural experiment to improve children’s dietary intake through school gardens. None of the schools in our sample had school feeding programs. With permission from school principals, we invited families of children attending elementary schools in our two target villages to attend a dinner and information session about the study. To participate, children needed to be students attending school, regardless of age. We did not employ any other inclusion or exclusion criteria. Parents provided written, informed consent, and children provided verbal assent to participate in the study. The Duke University Institutional Review Board and the Kenya Medical Research Institute Scientific and Ethics Review Unit reviewed and approved this study.

### 2.2. Dietary Assessment

Trained data collectors administered a single 24 h recall to document all foods and beverages consumed by children. We used Nutrition Data System for Research software version 2014 (University of Minnesota, Minneapolis, MN, USA) to calculate detailed nutrient information for each child. To calculate WDDS, we firstly aggregated foods into 10 mutually exclusive groups based on FAO’s guideline, namely (1) grains, white roots and tubers, and plantains; (2) pulses (beans, peas, and lentils); (3) nuts and seeds; (4) dairy; (5) meat, poultry, and fish; (6) eggs; (7) dark green leafy vegetables; (8) other vitamin A-rich fruits and vegetables; (9) other vegetables; and (10) other fruits [5]. To exclude very small quantities, the scoring criteria provides 1 point for each food group when intake is ≥15 g (referred to as minimum intake in this paper); if not, the point is 0 [5]. This results in a potential WDDS range from 0 to 10 for each child. Additionally, for the purpose of latent class analysis, we classified consumption of each food group as low (intake ≤ 20th percentile), medium (intake from >20th to ≤80th percentile), or high (intake > 80th percentile) within children in each village [29]. For several food groups of which more than 75% of children reported 0 consumption, we classified consumption of these food groups as any (intake > 0 g) or no (intake = 0 g), instead of low, medium, or high [29]. 

### 2.3. Anthropometric Assessment

Trained data collectors measured children’s height, weight, and mid-upper arm circumference (MUAC) using standard techniques [31,32]. We then calculated body mass index (BMI) as weight in kilograms divided by height in meters squared. 

### 2.4. Other Measures

We collected information from children on their birth year (most children did not know their specific birth date), gender, and orphan status (in village A only; we added the question after data collection had been completed in village B) via an in-person interview and questionnaire. 

### 2.5. Statistical Analysis

We first compared the distribution of WDDS component scores, as well as the prevalence of minimum intake of each food group, among children in the two villages. Next, we conducted the latent class analysis to assign individuals to their most likely dietary patterns based on their dietary intake data (i.e., 24 h recall), separately for children by village. We used indicators of consumption levels (i.e., low, medium, and high; or, any and no) of each food group as input variables in the latent class analysis. Simultaneously, we included age in years (continuous), gender (male, female), orphan status (children in village A only: yes, no), BMI (continuous), and total energy intake (continuous) as covariates in the latent class models to calculate whether these characteristics were associated with different odds of being classified into various dietary patterns. This procedure allowed us to examine the effects of covariates as part of the latent class analysis and is less prone to biased estimation of uncertainty [33]. We fitted models with two to seven latent classes and determined the best number of patterns to fit based on the Bayesian information criterion (BIC) and Akaike information criterion (AIC). We performed latent class analysis using the *poLCA* package [33] and all other analyses in R software version 3.6.3 (R Foundation for Statistical Computing) with a significance level of 0.05.

## 3. Results

Among the 1902 children who participated in the study, 759 (39.9%) were from village A and 1143 (60.1%) were from village B. The proportions of male and female children were comparable in the two villages (Table 1). Compared to children in village B, children in village A were, on average, slightly younger (mean age (standard deviation, SD): 11.6 (2.2) years versus 12.6 (2.2) years), but had a similar BMI (mean (SD) BMI: 17.3 (2.2) kg/m^2^ versus 17.5 (2.2) kg/m^2^) and similar MUAC (mean (SD) MUAC: 21.2 (2.9) cm versus 21.0 (2.8) cm). A higher proportion of children reported not going to bed hungry the night prior to assessment in village A (99.1%), compared to village B (92.6%). Just over one-third of the children in village A (*n* = 265, 34.9%) were orphans; we did not collect this information for village B. 

### 3.1. Dietary Diversity

We conducted dietary assessments in October 2013 for village A and July 2013 for village B. Children in village A consumed a mean (SD) of 966.1 (447.8) calories per day, compared to children in village B, who consumed a mean (SD) of 726.0 (414.3) calories per day (Table 2). The mean percentage of calories from fat, carbohydrates, and protein was 19.4%, 65.1%, and 15.5%, respectively, in village A, and 17.0%, 62.3%, and 20.6%, respectively, in village B. The mean (SD) WDDS in children in village A was 4.1 (1.4), compared to 2.6 (0.9) in village B. The prevalence of minimum intake (i.e., ≥15 g) was higher in village A than in village B for most of the food groups (Figure 1), with the exception of pulses (18.7% versus 25.6%), and meat, poultry, and fish (60.0% versus 77.2%). Almost all children (99.1% in village A and 95.3% in village B) consumed minimal grains, white roots and tubers, and plantains. Very few children consumed minimal nuts and seeds (0.9% in village A and 0.4% in village B) or dairy (3.7% in village A and 0.7% in village B). 

### 3.2. Dietary Patterns among Children in Village A

A total of 751 children from village A contributed to the latent class analysis. We dichotomized (i.e., any or no consumption) the following food groups due to the high prevalence of no intake: pulses; nuts and seeds; eggs; and other vitamin A-rich fruits and vegetables. We used three categories (i.e., low, medium, or high consumption) for all other food groups (Appendix A). Based on the BIC and AIC (Appendix A), we chose the model with three patterns as the final model for classifying dietary patterns among children in village A. The three dietary patterns were distinguished by food diversity (Figure 2, upper panel). We found that 28.5% (*n* = 214) of the children in village A were in the most diverse pattern, 33.8% (*n* = 254) in the moderately diverse pattern, and 37.7% (*n* = 283) in the least diverse pattern. The most diverse pattern was characterized by high consumption of grains, white roots and tubers, and plantains; dairy; meat, poultry, and fish; and other vegetables. The moderate and least diverse patterns differed in the consumption of other vitamin A-rich fruits and vegetables, as well as other fruits. We observed high consumption in the moderately diverse pattern, whereas low consumption was seen in the least diverse pattern. We found that age was not associated with dietary patterns in children in village A (Table 3). Being male, not being an orphan, and having a higher BMI were all associated with higher odds of being in a less diverse pattern after adjustment (Table 3). For example, the odds of being in the least versus the most diverse pattern was 5.66 times (95% CI: 4.58, 6.98) higher in male than in female children. Children with higher caloric intake had higher odds of being in a more diverse pattern after adjustment: for example, a 100 kcal/d increment in energy intake was associated with an odds ratio (OR) of 0.33 (95% CI: 0.26, 0.43) of being in the least versus the most diverse pattern. 

### 3.3. Dietary Patterns among Children in Village B

A total of 1131 children from village B contributed to the latent class analysis. We dichotomized nuts and seeds; dairy; eggs; other vitamin A-rich fruits and vegetables; and other fruits. We used three categories for all other food groups (Appendix A). We chose the model with three patterns as the final model. The most diverse dietary pattern (*n* = 654, 57.8%) was characterized by high consumption of grains, white roots and tubers, and plantains; pulses; meat, poultry, and fish; and other vegetables (Figure 2, bottom panel). The moderately diverse pattern (*n* = 331, 29.3%) was characterized by medium consumption of meat, poultry, and fish, and other vegetables. The least diverse pattern (*n* = 146, 12.9%) was characterized by high consumption of dark green leafy vegetables, but low consumption of meat, poultry, and fish. Age and gender were not associated with dietary patterns in children in village B (Table 3). Similar to children in village A, children with a higher BMI had higher odds of being in a less diverse dietary pattern (OR: 1.50, 95% CI: 1.15, 1.96 for being in the moderately diverse pattern; OR: 1.71, 95% CI: 1.29, 2.26 for being in the least diverse pattern), whereas children with higher caloric intake had higher odds of being in a more diverse dietary pattern after adjustment (Table 3).

## 4. Discussion

Our study is among the first to assess dietary diversity and characterize dietary patterns in a large cohort of school-aged children in rural Kenya. In 1902 children from two rural villages in western Kenya on Lake Victoria, we found that children in both villages had low dietary diversity and inadequate micronutrient intake, as reflected by a low WDDS. In particular, children reported consuming insufficient nuts and seeds, dairy, and eggs. We also identified three dietary patterns that were distinguished by food consumption levels and dietary diversity within the population in each village. Additionally, we found that children who were male, had a higher BMI, and lower daily caloric intake were more likely to have a less diverse dietary pattern. 

Our study adds to the literature base by using the 10-food-group-based WDDS to quantify dietary diversity. It has also been suggested as an indicator for micronutrient adequacy at the population level by the FAO [5]. The existing dietary diversity scores vary by numbers of food groups used for calculation, including 7, 8, 9, 10, or 12 food groups. The 10-food-group-based-WDDS used in our study has shown better performance in quantifying micronutrient adequacy than scores based on 7 or 9 food groups in women at reproductive age as well as in children of both sexes aged 4 to 8 years [6,7]. The dietary diversity score based on 12 food groups has been argued to be less reflective of micronutrient adequacy but a proxy for household-level food security [5]. Additionally, the Minimum Dietary Diversity for Women (MDD-W) is another indicator of micronutrient adequacy. It is based on the same 10 food groups used for the WDDS, but is dichotomized at the cut-off point of 5 [34]. We did not use the MDD-W in our study because it has been validated among women at reproductive age and is not recommended for use in other age groups or with males [34].

Our study provides the first line of evidence that school-aged children in rural areas in western Kenya had low dietary diversity and poor micronutrient adequacy, as reflected by the low WDDS calculated for 10 food groups. This is not unexpected given known low agricultural yields in the study area (Migori County) due to factors including the lack of irrigated acreage and the continued use of poor agricultural practices [35]. Previous studies that aimed to evaluate dietary diversity in Kenyan children have focused on young children under 5 years and used dietary diversity scores based on seven or nine food groups [8,9,10,11]. Studies in school-aged children in other African contexts, including in Ethiopia, Uganda, Malawi, Burkina Faso, Ghana, Nigeria, and Tanzania, have reported dietary diversity scores that were relatively comparable with scores in our study [12,13,14,15]. These studies also used inconsistent dietary diversity scores varying by numbers of food groups [12,13,14,15]. Based on 8 food groups, the mean (SD) score was 5.8 (1.1) in children aged 10 to 14 years in schools with feeding programs and 3.5 (0.7) in schools without feeding programs in southern Ethiopia [14]. As noted previously, neither of the schools in our sample had feeding programs. Based on 9 food groups, the median (25th percentile, 75th percentile) score was 3 (2, 3) among girls aged 10 to 18 years in child care institutions and 3 (3, 4) among girls in boarding schools in Uganda [13]. Based on 12 food groups, children aged 10 to 19 years from 6 countries (not including Kenya) in sub-Saharan Africa reported a mean score ranging from 5.1 to 7.8 [15]; and children aged 8 to 18 years from 4 villages in Malawi reported a mean score ranging from 2.9 to 5.5 [12]. 

Our study identified dietary patterns in school-aged children and joins a small body of literature that assessed dietary characteristics in Kenya. Previous studies were conducted either in young children [22,24], adults [21,25,26], or households [23,27,36]. They also did not identify dietary patterns but instead reported consumption frequency or prevalence of selected foods. A recent systematic review and meta-analysis on studies focusing on dietary behaviors in urban Kenya found substantial heterogeneity with regard to types of food consumed, as well as dietary practices across different studies [37]. Similarly, we observed some differences in food group consumption and dietary patterns between children in village A and B. Children in village A had a substantially higher prevalence of minimum consumption of almost all food groups, compared to children in village B—with the exception of pulses, meat, poultry, and fish. These differences could be due to distinctive geographic locations; village A is located in a more fertile area conducive to farming, whereas village B is located in a more arid fishing community. The geographic differences are reflected in the higher reported consumption of fruits and vegetables in village A and fish consumption in village B. Moreover, in village B, we conducted assessments in July at the end of the long rainy season when the harvest had yet to begin [38]. In village A, we conducted assessments in October, when the rainy season had ended and harvest had begun [39]. This could also partially explain the overall better diet intake profile in village A than village B. 

Geography and seasonality could also help explain the differences in dietary patterns observed between the two villages in our study. In village A, the dietary patterns were relatively equally distributed among children. There was a slightly more diverse and sufficient food supply available in village A—mainly due to a more favorable farming environment. In village B, the three dietary patterns had quite different proportions. The region was less conducive to farming and was instead a fishing community. Therefore, it is not surprising that all, but especially the most diverse pattern, had relatively high fish consumption. 

We also observed differences in dietary patterns that were associated with gender, BMI, and energy intake. We found that males were more likely to have a less diverse dietary pattern than females. We were only able to find limited studies that examined gender differences in dietary diversity in school-aged children in Africa. A recent study examined food consumption in over 7000 children aged 10 to 19 years from 6 countries in sub-Saharan Africa, including Burkina Faso, Ethiopia, Ghana, Nigeria, Uganda, and Tanzania [15]. Consistent with our study, they found that female children had a higher probability of consuming meats, eggs, and fish, but not cereals, white roots and tubers, vegetables, fruits, legumes, nuts and seeds, or dairy [15]. However, notable heterogeneities existed across countries. For example, no gender differences were observed in children from Uganda or Tanzania [15]. As for associations of dietary diversity with BMI and energy intake, positive, null, or negative findings have all been reported by similar proportions of studies conducted in a variety of settings and populations according to a systematic scoping review [3]. The inconsistent findings from these studies could be partially due to the fact that they were conducted in different study populations, used different types of dietary diversity indicators, and adjusted for different sets of covariates. Our findings that in school-aged children in rural Kenya, a dietary pattern with higher dietary diversity was associated with a lower BMI but higher energy intake with adjustment for these and multiple other factors, should be further investigated. 

According to the World Health Organization (WHO), currently there is no dietary recommendation of global utility available for children and adolescents [40]. The American Heart Association (AHA) recommends a daily intake of 1600 kcal for females and 1800 kcal for males aged 9 to 13 years [41]. The children in our study consumed fewer daily calories (966.1 kcal in village A and 726 kcal in village B) than the AHA recommendation, which also fell below the median daily energy intake (1700 kcal) of adolescents ages 9 to 13 years in rural South Africa reported by a review of 67 studies [42]. Additionally, AHA recommends 3 cups (about 750 g) of milk or dairy per day, whereas only 3.7% of children in village A and 0.7% of children in village B consumed more than 15 g of dairy per day. With regard to the percentages of total energy from carbohydrates, fat, and protein, they fell within the population nutrient intake goals set by the WHO, which are 55–75% from carbohydrates, 15–30% from fat, and 10–15% from protein [43]. Taken together, these findings emphasize the need to increase children’s dietary intake in rural Kenya in order to maintain healthy growth and development.

Although our study contributes to the growing body of literature in this area, it also has limitations. First, we used a single (rather than multiple) 24 h recall to measure diet, which could have resulted in random measurement error if one day did not adequately reflect usual intake. Nevertheless, the WDDS was designed to be used with a single 24 h recall to measure dietary diversity and reflect population-level micronutrient adequacy. Second, seasonal differences in food availability in African populations have been well-documented [44,45,46], but our study was not designed to capture these changes. Third, our population is a convenience sample of school-aged children in one county (Migori) and is therefore not reflective of populations of different ages, or those residing in other geographic areas in or outside of Kenya. Fourth, we were not able to examine whether sociodemographic characteristics, such as household income or family composition, were associated with dietary patterns due to the limited characteristics collected. We also did not have specific data on child age because most of the children did not know their birth month or sometimes year. 

Despite limitations, our study enriched the existing research on school-aged children’s dietary intake in rural Kenya by having a large sample size and including both females and males. We also used the 10-food-group-based WDDS to quantify children’s dietary diversity and overall micronutrient adequacy, which outperforms indicators based on other numbers of food groups used by previous studies [5,6,7]. We joined a small but growing body of literature using latent class analysis to summarize dietary patterns. We were able to characterize children based on the whole of their reported dietary intake, directly adjusting for important covariates. We also evaluated associations between these factors and dietary patterns. Lastly, stratifying by the village was appropriate because the covariate effects (e.g., betas) likely varied across the two geographic locations. Combining children from both villages into one model would require the inclusion of multiple two-way interactions in the model to allow for this degree of effect modification, which would be challenging to estimate. A stratified approach, therefore, offers a more parsimonious and interpretable analysis to accommodate the full interaction model. 

## 5. Conclusions

Our study addressed the important but under-studied issue of dietary intake in school-aged children in western Kenya. We found that, generally, children in two rural villages in western Kenya had low dietary diversity and inadequate micronutrient consumption. We characterized three dietary patterns that differed by dietary diversity within the population in each village, and found that being male, having a higher BMI, and having lower caloric intake were associated with a less diverse dietary pattern. Our findings emphasize the need for additional resources to improve dietary intake among school-aged children, especially in rural settings in Kenya. However, more studies with additional dietary assessments conducted at multiple time points across the year are needed to assess the potential impact of seasonality on dietary diversity and dietary patterns. Additionally, we recommend future studies collect a broader range of demographic and socioeconomic characteristics in order to explore key influencers in dietary habits in school-aged children in western Kenya. 

## Figures and Tables

**Figure 1 ijerph-19-09130-f001:**
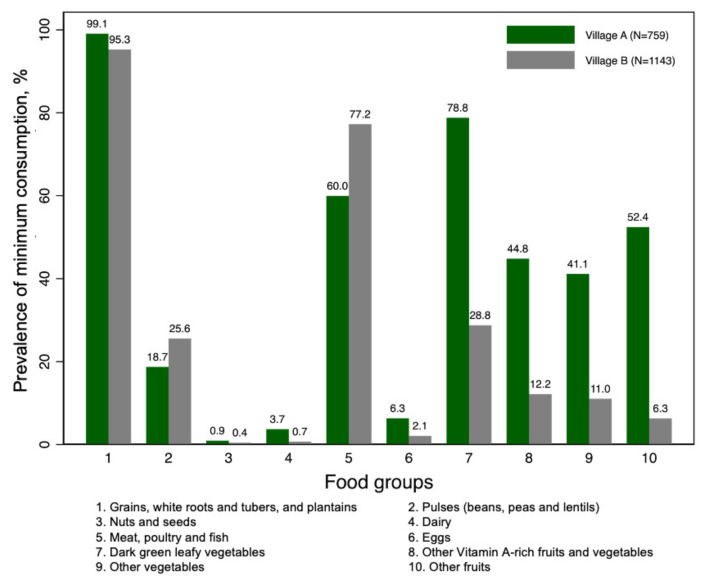
Prevalence of minimum consumption of food groups in children in two villages in western Kenya. Minimum consumption is defined as food group intake ≥ 15 g.

**Figure 2 ijerph-19-09130-f002:**
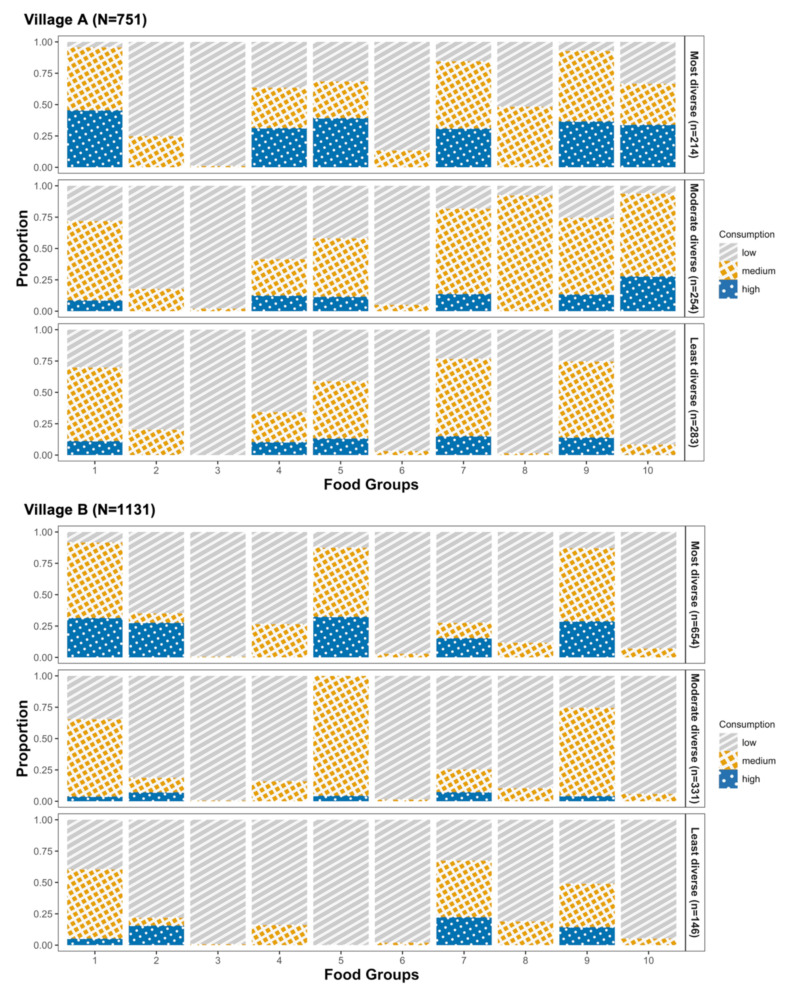
Dietary patterns in children in village A (**upper**) and village B (**lower**) in western Kenya. The horizontal axis corresponds to food groups: (1) grains, white roots and tubers, and plantains; (2) pulses (beans, peas, and lentils); (3) nuts and seeds; (4) dairy; (5) meat, poultry, and fish; (6) eggs; (7) dark green leafy vegetables; (8) other Vitamin A-rich fruits and vegetables; (9) other vegetables; and (10) other fruits. The vertical axis is the prevalence of children consuming a low, medium, and high amount of each food group within each dietary pattern.

**Table 1 ijerph-19-09130-t001:** Sociodemographic and anthropometric characteristics of children in two villages in western Kenya.

	Village A (*n* = 759)	Village B (*n* = 1143)
	Percentage (Number)	
Gender, male	53.6 (407)	52.2 (596)
Did not go to bed hungry last night	99.1 (752)	92.6 (1058)
Not orphan	65.1 (494)	--
	Mean (SD)	
Age, years	11.6 (2.2)	12.6 (2.2)
Height, cm	147.0 (12.4)	148.5 (13.0)
Weight, kg	38.2 (10.3)	39.3 (10.6)
BMI, kg/m^2^	17.3 (2.2)	17.5 (2.2)
MUAC, cm	21.2 (2.9)	21.0 (2.8)

Abbreviations: BMI, body mass index; MUAC, mid-upper arm circumference; SD, standard deviation.

**Table 2 ijerph-19-09130-t002:** Dietary intake summaries of children in two villages in western Kenya.

	Village A (*n* = 759)	Village B (*n* = 1143)
	Mean (SD)	
Total daily calories	966.1 (447.8)	726.0 (414.3)
Percentage calories from fat, %	19.4 (9.5)	17.0 (9.5)
Percentage calories from carbohydrates, %	65.1 (13.3)	62.3 (15.6)
Percentage calories from protein, %	15.5 (9.8)	20.6 (12.5)
WDDS	4.1 (1.4)	2.6 (0.9)
	Percentage (number)	
WDDS component scores		
0	0.8 (6)	0.4 (5)
1	0.5 (4)	6.6 (76)
2	11.9 (90)	43.0 (491)
3	23.5 (178)	36.5 (417)
4	24.5 (186)	10.1 (116)
5	23.7 (180)	3.0 (34)
6	11.6 (88)	0.3 (4)
7·	3.4 (26)	0.0
8	0.1 (1)	0.0
9	0.0	0.0
10	0.0	0.0

Abbreviations: WDDS, women’s dietary diversity score; SD, standard deviation.

**Table 3 ijerph-19-09130-t003:** Odds Ratios and 95% Confidence Intervals (CI) between children’s characteristics and dietary patterns in two villages in western Kenya.

	Odds Ratio (95% CI)		
	Most Diverse	Moderately Diverse	Least Diverse
**Village A (*n* = 751)**	*n* = 214	*n* = 254	*n* = 283
Age, years	reference	1.13 (0.87, 1.49)	1.15 (0.88, 1.50)
Gender ^a^	reference	2.44 (1.89, 3.15)	5.66 (4.58, 6.98)
Orphan status ^b^	reference	4.19 (2.66, 6.59)	3.58 (2.50, 5.13)
BMI, kg/m^2^	reference	1.50 (1.20, 1.86)	1.57 (1.26, 1.96)
Per 100 kcal/d energy	reference	0.42 (0.32, 0.55)	0.33 (0.26, 0.43)
**Village B (*n* = 1131)**	*n* = 654	*n* = 331	*n* = 146
Age, years	reference	1.13 (0.91, 1.41)	1.00 (0.78, 1.27)
Gender ^a^	reference	0.95 (0.39, 2.28)	1.34 (0.52, 3.47)
BMI, kg/m^2^	reference	1.50 (1.15, 1.96)	1.71 (1.29, 2.26)
Per 100 kcal/d energy	reference	0.22 (0.11, 0.41)	0.14 (0.07, 0.28)

^a^ Comparing male versus female. ^b^ We collected data on orphan status in village A only, comparing non-orphans versus orphans.

## Data Availability

Data described in the manuscript, code book, and analytic code will be made available upon request pending appropriate institutional and ethical agreements.

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
