# Peer review of "Dietary Diversity and Dietary Patterns in School-Aged Children in Western Kenya: A Latent Class Analysis"

_ijerph, 2022, doi:10.3390/ijerph19159130_

Round 1

Reviewer 1 Report

The authors' manuscript addresses dietary patterns among school-aged children in Western Kenya.  This is an exceptionally well-written manuscript.  Kudos to the authors from Duke and Johns Hopkins for addressing this important issue in Western Kenya.  

Figure 2:  Perhaps the authors could provide a variety of cross-hatches or patterns for the presented levels of consumption; the B/W images make it difficult to discern those levels. 

While the authors comment on AHA recommendations (pg 9), are those recommendations really practical across the globe, particularly among resource-challenged populations?  This reviewer has worked among resource-challenged populations on every continent.  It's imperative that dietary recommendations consider the dynamics of culture, tradition, and agricultural practices/products as well as import/export economics.

The authors' punch line (pg 9, L 328), their work markedly enriched our understanding of school-aged children in rural Kenya.  As the authors state (L343), children in areas, such as Western Kenya, should be studied further and should receive additional assistance from agricultural practices and post-harvest management (sometimes losses exceed 50%). 

Excellent choices of references, particularly 12-15.
